# Patterns of Muscle-Related Risk Factors for Sarcopenia in Older Mexican Women

**DOI:** 10.3390/ijerph191610239

**Published:** 2022-08-18

**Authors:** María Fernanda Carrillo-Vega, Mario Ulises Pérez-Zepeda, Guillermo Salinas-Escudero, Carmen García-Peña, Edward Daniel Reyes-Ramírez, María Claudia Espinel-Bermúdez, Sergio Sánchez-García, Lorena Parra-Rodríguez

**Affiliations:** 1Instituto Nacional de Geriatría, Dirección de Investigación, Av. Contreras 428, Ciudad de México 10200, Mexico; 2Centro de Investigación en Ciencias de la Salud (CICSA), Universidad Anáhuac México Campus NorteFCS, Huixquilucan 52786, Mexico; 3Hospital Infantil de Mexico Federico Gómez, Centro de Estudios Económicos y Sociales en Salud, Calle Doctor Márquez 162, Ciudad de Mexico 06720, Mexico; 4Instituto Mexicano del Seguro Social, Centro Mexico Nacional de Occidente, Unidad Médica de Alta Especialidad Hospital de Especialidades, Unidad de Investigación Biomédica 02 y División de Investigación en Salud, Av. Belisario Domínguez 1000, Guadalajara 44340, Mexico; 5Instituto Mexicano del Seguro Social, Centro Médico Nacional Siglo XXI, Unidad de Investigación en Epidemiología y Servicios de Salud, Área de Envejecimiento, Av. Cuauhtémoc 330, Ciudad de México 06720, Mexico

**Keywords:** aged, body composition, sarcopenia, functional decline, physical performance tests

## Abstract

Early detriment in the muscle mass quantity, quality, and functionality, determined by calf circumference (CC), phase angle (PA), gait time (GT), and grip strength (GSt), may be considered a risk factor for sarcopenia. Patterns derived from these parameters could timely identify an early stage of this disease. Thus, the present work aims to identify those patterns of muscle-related parameters and their association with sarcopenia in a cohort of older Mexican women with neural network analysis. Methods: Information from the functional decline patterns at the end of life, related factors, and associated costs study was used. A self-organizing map was used to analyze the information. A SOM is an unsupervised machine learning technique that projects input variables on a low-dimensional hexagonal grid that can be effectively utilized to visualize and explore properties of the data allowing to cluster individuals with similar age, GT, GSt, CC, and PA. An unadjusted logistic regression model assessed the probability of having sarcopenia given a particular cluster. Results: 250 women were evaluated. Mean age was 68.54 ± 5.99, sarcopenia was present in 31 (12.4%). Clusters 1 and 2 had similar GT, GSt, and CC values. Moreover, in cluster 1, women were older with higher PA values (*p* < 0.001). From cluster 3 upward, there is a trend of worse scores for every variable. Moreover, 100% of the participants in cluster 6 have sarcopenia (*p* < 0.001). Women in clusters 4 and 5 were 19.29 and 90 respectively, times more likely to develop sarcopenia than those from cluster 2 (*p* < 0.01). Conclusions: The joint use of age, GSt, GT, CC, and PA is strongly associated with the probability women have of presenting sarcopenia.

## 1. Introduction

Sarcopenia, the progressive and generalized loss of skeletal muscle mass and function, leads to functional decline, disability, falls, fractures, frailty, hospitalization, higher hospital length of stay, increased healthcare costs, and mortality [1,2,3]. It denotes a significant impact on quality of life and increased health services use [4,5,6].

The deterioration that precedes sarcopenia has a dynamic, non-linear nature, whose patterns differ among individuals [7], but these differences are especially significant between men and women [8,9]. This is also apparent when studying the differences according to the sex of physical performance tests [10,11,12] and is commonly attributed to body composition. For example, it has been demonstrated that higher adiposity in women decreases scores on physical performance tests [13]. Moreover, muscle fat infiltration is related to lower strength and overall physical function [14,15,16]. In addition to those factors related to body composition, other causes have been described, such as poorer health, a higher burden of chronic diseases, higher life expectancy and diminished testosterone levels [14,17,18].

Evidence suggests that a decline in the muscle mass quantity, quality, and functionality [19,20,21] may herald sarcopenia in its full expression. Standardized methods for the assessment of these alterations are widely known. However, in some contexts techniques such as dual energy X-ray absorptiometry (DXA) are not available due to their high cost and poor portability. Calf circumference (CC) as an indicator of the muscle mass quantity, phase angle (PA) for the assessment of quality, and gait time (GT) and grip strength (GSt) for functionality, are inexpensive tools that do not require the use of highly sophisticated equipment. Additionally, they are extensively used both in the clinical setting and in geriatric research. Early changes in these parameters may be considered risk factors for sarcopenia [22,23,24,25,26,27,28,29,30,31]. Therefore, we hypothesize that patterns derived from these parameters could timely identify an early stage of sarcopenia. Thus, the present work aims to identify those patterns of muscle-related parameters and their association with sarcopenia in a cohort of older Mexican women, with neural network analysis.

## 2. Materials and Methods

### 2.1. Study Population and Design

The “functional decline patterns at the end of life, related factors, and associated costs study (FDP)” is a cohort of subjects aged 60 years or older. This study is an arm of the Cohort of Obesity, Sarcopenia, and Frailty of Older Mexican Adults (COSFOMA), which has been described in detail elsewhere [32,33,34]; in brief, this cohort belongs to the most extensive Mexican health subsystem: the Mexican Social Security Institute (IMSS). The FDP aims to determine functional decline patterns, their determinants, and their associated costs. To reach this goal, older adults were measured at baseline (during 2016), ≈12 and ≈24 months after (during 2017 and 2018). Biological, clinical, psychological, and sociodemographic variables were investigated by direct interview. Anthropometry was determined following a validated methodology and by previously standardized personnel.

As per those particular features of women previously described, only women are included in this work. Figure 1 shows the sample size flowchart for the conformation of the final sample.

### 2.2. Variables

#### 2.2.1. Muscle-Related Risk Factors

CC (cm), PA (degrees), GT (seconds), and GSt (Kg) were included in the patterns. CC was evaluated as part of the anthropometric evaluation in agreement with international criteria [35]. An RJL Systems device was used to conduct a bioimpedance analysis with an operating frequency of 50 kHz at 800 μA [36]. The data on Resistance (R) and Reactance (Xc) obtained by this test were used to calculate the PA as the arctan Xc/R 180/πA. GT was considered as the time in seconds it takes the individual to walk a distance of 15 feet (4.5 m), stratified by sex and height. A Takei hand dynamometer (Takei TKK 5001, Takei Scientific Instruments Co. Ltd., Tokyo, Japan) was used to measure grip strength; three measurements were taken from each side, and the highest was considered. Age was included to study its effect on the conformation of the patterns. It was asked in the interview and corroborated by the date of birth reported by the participant.

#### 2.2.2. Sarcopenia

Sarcopenia was defined using the revised criterion of the European Working Group on Sarcopenia in Older People (EWGSOP2) [8]. The diagnosis was established when low muscle strength, measured by grip strength, and low muscle mass, measured by appendicular skeletal muscle mass (ASMM) values. For the present work, the data from the bioimpedance analysis were used to estimate the ASMM by the equation that Rangel Peniche et al. validated on the Mexican population [26].

Other variables included to characterize the sample were: body mass index (BMI), smoking status, chronic diseases self-report, ≥3 medications, falls in the previous year, hospitalizations during the previous year, depression (Center for Epidemiologic Studies

Depression Scale-Revised, CESD-R) [37,38], Activities of Daily Life (ADLs-Barthel Index) [39], Instrumental Activities of Daily Life (IADLs-Lawton and Brody index) [40], malnutrition risk (MNA-SF) [41], and quality of life (SF-36) [42].

### 2.3. Analytic Plan

A descriptive analysis was performed for the individuals at baseline. Continuous variables are presented as means and standard deviations (SD), and categorical variables are expressed as numbers and percentages.

The neural network analysis was performed with a self-organizing map (SOM) to compare and group women with similar age, GT, GSt, CC, and PA. For this purpose, the five variables were standardized to be dimensionless and have the same scale, with a mean of zero and a standard deviation of one. SOM is a powerful unsupervised machine learning technique used to produce a low-dimensional representation of a higher dimensional data set while preserving the topological structure of the data [43]. It projects input variables on a low-dimensional hexagonal grid that can be effectively utilized to visualize and explore properties of the data. When the number of SOM units is large, to facilitate quantitative analysis of the map and the data, similar units are grouped, i.e., clustered [44]. Several mathematical techniques have been developed to analyze multivariate data including principal component analysis, K-means and other clustering and dimensional reduction methods. Self-organizing neural networks have proved to be one of the best algorithms that is useful to automatically analyze, characterize, and classify entities with multivariate data. Furthermore, the main advantage of the SOM over these other techniques is that it visually represents data and the knowledge obtained by the mathematical computations in a low-dimensional space that fully contains high-dimensional information [45,46,47,48,49,50]. Additionally, it has been reported that the joint use of SOM with other traditional regression models provides promising results in the medicine field, particularly when complex and non-linear are modelled [50].

The software tool used in this study can be obtained freely from the Nonlinear Dynamics Laboratory Web Page [51]. Two visualization sceneries were used: (1) a clusters’ map that visually depicts the identified groups and (2) a set of maps (one map for each variable) that allow us to characterize the profiles of the participants.

Each identified cluster was labeled with a number and colored with a random color. Indicators maps are colored according to a chromatic scale, ranging from the highest values in green, the lowest in red, and yellow for intermediate values. To determine the clusters’ characteristics, we look up the colors in the same zone but on the dimension’s maps. The spatial distribution of clusters also obeys profile similarity. Thus, two adjacent clusters are more similar than those that are not adjacent.

A Shapiro-Wilk test was used to assess the normality of the distribution of the variables in each cluster (age, gait time, grip strength, calf circumference, and PA. The differences between means were tested using an ANOVA test for the variables that resulted normal and a Kruskal–Wallis test for the variables that were not normally distributed. A 𝜒^2^ test was used to determine if having sarcopenia was related to the cluster assignment. An unadjusted logistic regression model was run to assess the probability of having sarcopenia. For this purpose, individuals in clusters 1, 2, and 3 were collapsed in one group and participants in clusters 4, 5, and 6 were gathered in a second group.

### 2.4. Ethics

This study was conducted in accordance with the Declaration of Helsinki. The protocol was reviewed and approved by the National Committee of Scientific Investigation of the IMSS (Registry Number: R-2017-785-018) and the Research and Research Ethics Committees of the National Institute of Geriatrics (Registry Number: DI-PI-010/2016).

## 3. Results

A total of 250 women with a mean age of 68.54 ± 5.99 were evaluated. The values of the muscle-related risk factors (gait time, grip strength, calf circumference, and PA) were 5.11 ± 3.25 s, 20.98 ± 5.36 kg, 34.15 ± 3.14 cm, and 6.26 ± 1.05 grades, respectively. Sarcopenia was present in 31 (12.4%) older women. Other characteristics of the sample are described in Table 1.

Figure 2 presents the results of the neural network analysis. At the top left, variable maps colored according to a chromatic scale can be found. Younger women with better GT and GSt, are in the lower left quadrant, while older women with worse values in these variables can be seen in the right upper quadrant. Concerning CC, a smaller number of women in the lower right quadrant have greater values, while the majority have lower values represented in red. Regarding the PA, in the lower-left quadrant, a small number of women with high values is observed, with a decrease in the same parameter in women who approach the upper right quadrant. At the top right of the figure the clusters’ map that contains the information of the variables analyzed together can be found. Six patterns were created from the visualization of the variable maps and the research group’s expertise. The further the clusters are from each other, the more different the individuals among them will be. In this map, cluster 1 contains individuals with better values in most parameters; cluster 6 contains the women with greater impairment. Cluster 3, for its part, gathers women with similar conditions to those of the two adjacent clusters.

The descriptive analysis of the variables for each cluster is presented in Table 2. Clusters 1, 2, and 3 gather the youngest participants with best physical performance and body composition, while clusters 4–6 contain the individuals with worst results in all variables. Clusters 1 and 2 have similar gait time, grip strength, and calf circumference values, but in cluster 1, women are older with higher PA values (*p* < 0.001). From cluster 4 upward, there is a trend of worse scores for every variable, ending with 100% of the participants in cluster 6 having sarcopenia (*p* < 0.001).

The differences between the means of the clusters were tested using a Kruskal–Wallis test for non-normal variables (age, gait time, calf circumference and phase angle) and a one-way ANOVA for grip strength.

The whole logistic regression model is presented in Table 3. It can be said that women in clusters 4, 5, or 6 are 20.02 times more likely to develop sarcopenia than women from clusters 1, 2, or 3 (*p* < 0.01).

## 4. Discussion

To the best of our knowledge, the present study is the first to use a neural network analysis technique to show early patterns of muscle-related risk factors for sarcopenia in a group of older women. Our findings revealed six patterns according to the variables’ proximity. Women with higher scores in the muscle-related risk factors variables are grouped in clusters 1 and 2, while clusters 5 and 6 bring together those women with more significant impairment. Clusters 2–4 gather 88% of the sample with similar characteristics, observing moderately good values with a tendency to deteriorate as they approach the upper right frame of the map.

The resulting clusters align with the previous evidence on the sarcopenia components and pathway [52]. It can be seen that the prevalence of sarcopenia increases as the assigned cluster number progresses. Further, the logistic regression model results confirm that belonging to a specific pattern is associated with the risk of having sarcopenia. According to this information, network analysis can be an effective technique for the timely identification of groups of older women at risk of sarcopenia.

For the conformation of patterns, we used CC and PA in addition to GSt and GT, which are part of the comprehensive geriatric assessment. CC and PA are minimally invasive, easy to apply, portable, and cheaper than other tests, attractive elements for geriatric evaluation. CC is a widely used assessment for exploring muscle mass in clinical settings. On the other hand, PA is a parameter closely related to muscle quality [53]. Current evidence has been published on the correlation between these parameters and other components of the sarcopenia diagnosis criteria [20,54,55,56,57]. There is solid evidence of the four variables as independent risk factors for developing sarcopenia, functional decline, and dependence [58,59,60].

The results on PA are especially noteworthy since it is a parameter of growing interest in the clinical and research settings. In our sample, the mean value for PA in the cluster with the most significant impairment was 4.7°. Kilic et al. [61] reported the association of PA with muscle loss when values are below 4.55°. According to Kołodziej et al. [31], a value above 4.7° can identify pre-sarcopenia in women. This evidence suggests using this parameter, when available, during the comprehensive geriatric evaluation as a predictor of muscle mass and functionality.

It is not surprising to observe an increase in age as the clusters move closer to the upper right quadrant, the one with the worst profile. This result agrees with the postulate that age is the primary determinant of sarcopenia [62,63]. Moreover, our results add to what already exists on the higher prevalence of sarcopenia with advancing age [64,65].

Some limitations need to be declared. In women, depression and the hormonal profile are some risk factors for sarcopenia, functional decline, and dependence [66,67,68]. The logistic regression analysis did not use these and other relevant variables as adjustment variables. Notwithstanding, the present analysis was aimed at identifying the joint role of certain variables in determining risk profiles for sarcopenia via network analysis. Thus, adjustment variables were not a requisite. Second, although the ratios obtained using the logistic regression model were statistically significant, the results should be taken with caution due to the width of the confidence intervals. Nevertheless, when analyzed with the clustering map, these results point towards patterns of muscle-related risk factors for sarcopenia. Third, the sample size is affecting the statistical significance of the logistic model and the representativeness of the study. Moreover, we know that a larger sample size would increase the definition of the clusters. However, the statistical differences between the values in the different clusters allow us to conclude that the technique has the power to visualize a complex phenomenon at a given moment and to accurately identify subgroups of individuals with similar characteristics and specific needs.

It should be noted that despite its limitations, neural network analysis represents a relevant approach for health sciences for several reasons. One of them is that it does not use cut-off points for the variables of interest, which is convenient for populations with no specific cut-off points, as is the case of the Mexican population. In addition, compared to other statistical models in which the results variable is binary, neural network models visualize the combination of more than one continuous variable. These elements make neural network analysis a powerful tool for epidemiological and clinical settings.

## 5. Conclusions

Although there is no evidence of the joint use of age, GSt, GT, CC, and PA as risk factors for sarcopenia, and even less for its diagnosis and prognosis, our results point out that together they are strongly associated with the probability of women have of presenting sarcopenia. Our work is one of the first including neural network analysis as its primary technique [69]. This approach allows to identify specific groups of women in whom adequate interventions can slow, or even more, brake, sarcopenia.

## Figures and Tables

**Figure 1 ijerph-19-10239-f001:**
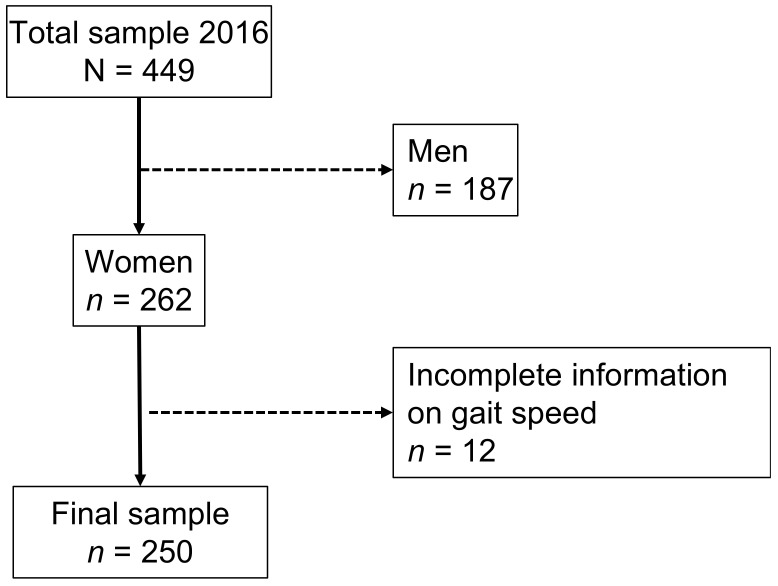
Sample size flowchart.

**Figure 2 ijerph-19-10239-f002:**
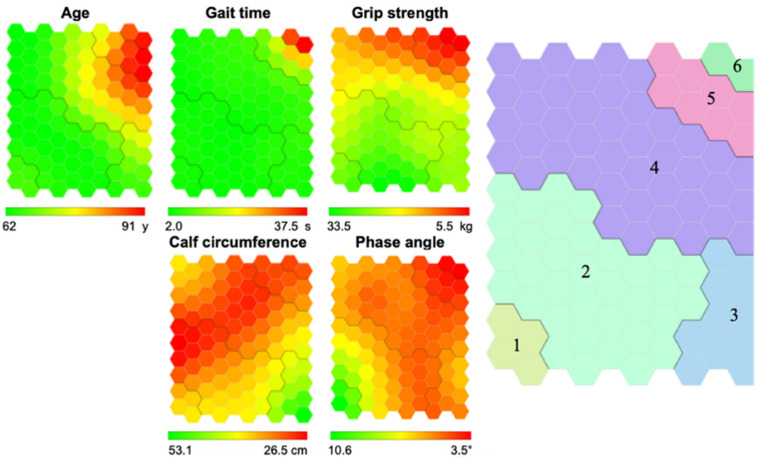
Variable maps, cluster map and characteristics of the six clusters. **Left**: Variable maps are colored according to a chromatic scale. The best physical performance and body composition values are colored in green (lower Age and higher Gait time, Grip strength, calf circumference, and phase angle), worst values are colored in red (higher Age and lower Gait time, Grip strength, Calf circumference, and Phase angle) and intermediate values are colored in yellow and orange. **Right**: Cluster Map obtained by the Neural Network Analysis. Colors were added randomly. Each identified cluster is labeled with a number as the test results worsen.

**Table 1 ijerph-19-10239-t001:** General characteristics of the sample.

	*n* = 250
Age, years	68.54 ± 5.99
Functional pattern	
Gait time, s	5.11 ± 3.25
Grip strength, kg	20.98 ± 5.36
Calf circumference, cm	34.15 ± 3.14
Phase angle, grades	6.26 ± 1.05
Weight, kg	65.32 ± 10.83
Height, cm	152.58 ± 5.99
BMI, kg/m^2^	28.05 ± 4.28
Sarcopenia, *n*(%)	31 (12.4)
Smoking, *n* (%)	99 (39.6)
Arthritis, *n* (%)	5 (2.0)
COPD, *n* (%)	1 (0.4)
Type 2 Diabetes, *n* (%)	56 (22.4)
Hypertension, *n* (%)	97 (38.8)
Heart Diseases, *n* (%)	90 (36.0)
Depression (CES-D 20 score ≥ 20), *n* (%)	42 (16.8)
Polypharmacy	119 (47.6)
Fell in the previous year, *n* (%)	72 (28.8)
Hospitalization in the previous year, *n* (%)	19 (7.60)
Activities of Daily Life, score	99.36 ± 3.07
Instrumental Activities of Daily Life, score	7.76 ± 0.83
Risk of malnutrition (MNA score < 24)	49 (19.6)
Quality of Life	
Physical functioning	80.64 ± 22.37
Role limitations due to physical health	74.20 ± 39.18
Role limitations due to emotional problems	81.47 ± 35.83
Energy/fatigue	65.50 ± 19.67
Emotional well-being	75.20 ± 18.65
Social functioning	87.20 ± 21.43
Pain	74.00 ± 25.61
General health	61.00 ± 19.19

**Table 2 ijerph-19-10239-t002:** Characteristics of the six clusters.

*n* = 250	Cluster 1 *n* = 11	Cluster 2 *n* = 91	Cluster 3 *n* = 27	Cluster 4 *n* = 102	Cluster 5 *n* = 16	Cluster 6 *n* = 3	*p* Value
Age, years	67.4 ± 4.2	64.6 ± 2.1	68.5 ± 3.9	70 ± 5.6	79.2 ± 6.4	83 ± 7.5	<0.001
Gait time, s	4.4 ± 1.3	4.3 ± 1	4.7 ± 1.1	4.8 ± 1.2	8.5 ± 2.4	30 ± 8.4	<0.001
Grip strength, kg	24.2 ± 4.9	24.4 ± 3.9	23.1 ± 2.8	18.6 ± 4	12.5 ± 4	9.5 ± 4.6	<0.001
Calf circumference, cm	35.2 ± 2.8	33.7 ± 2.6	39.4 ± 3.4	33.4 ± 2.2	32.6 ± 2.1	30.9 ± 1.3	<0.001
Phase angle, grades	9.3 ± 0.8	6.4 ± 0.8	6.4 ± 0.8	6 ± 0.7	5.2 ± 0.4	4.7 ± 1	<0.001
Sarcopenia, *n* (%)	0	1 (1.1)	1 (3.7)	18 (17.6)	8 (50.0)	3 (100)	<0.001

**Table 3 ijerph-19-10239-t003:** Logistic regression model for the risk of sarcopenia.

*n* = 209	Odds Ratio	Std. Err.	*p*	95% C.I.	
Belonging to clusters 1, 2, or 3 (ref)					
Belonging to clusters 4, 5, or 6	20.02	0.744	<0.001	4.66	86.00
Constant	0.02	0.713	<0.001		

## Data Availability

The data presented in this study are available on reasonable request from the corresponding author. The data are not publicly available due to privacy.

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
