# Peer review of "Patterns of Muscle-Related Risk Factors for Sarcopenia in Older Mexican Women"

_ijerph, 2022, doi:10.3390/ijerph191610239_

Round 1
Reviewer 1 Report
I thank the editor for the opportunity to do this review.
The topic is highly relevant, as the search for potential biomarkers to help prevent the development of sarcopenia in older adults remains a challenge for the scientific community.
On the other hand, using neural networks as a tool for data analysis may open the way to new possibilities in medicine.
I want to enumerate some of the comments below:
1. LINES 36-57: INTRODUCTION
Commentary: Could the authors justify in the introduction why these parameters (CC, PA, GSp, GSt) and not others?; and why is it better to use calf circumference and not appendicular skeletal muscle mass?
2. LINES 52-53: “Indeed, early changes in calf circumference (CC), phase angle (PA), gait speed (GSp), and grip strength (GSt) may be considered muscle-related risk factors for sarcopenia”
Commentary: Please insert a reference to this statement or provide further information to support it.
3. LINES: 77-88: 2.2.1. Muscle-related risk factors
Commentary: Could the authors please specify the units of measurement used (cm, m, m/s, Kg, N)?.
4. LINES 139 -143: “3. Results… This section may be divided by subheadings. It should provide a…”
Commentary: This explanation is unnecessary here. Could this paragraph be a drafting error?
5. LINES 146-147 and Table-1, Row 3: “gait speed .. s”
Commentary: if the variable is "speed," its results must be given in the appropriate units (m/s).
6. LINE 180: “clusters 1, 3, and 6 were eliminated as they had a small sample size…”
Commentary: This is confusing; cluster 5 has a smaller sample size than cluster 3.
7. LINES 202-205: … we used CC and … which are part of the commonly used tests for sarcopenia…
Commentary: The use of calf circumference as an assessment in sarcopenia is limited in the lack of other assessment tools. I quote text from EWGSOP2(1):
“Although anthropometry is sometimes used to reflect nutritional status in older adults, it is not a good measure of muscle mass [78]. Calf circumference has been shown to predict performance and survival in older people (cut-off point <31 cm) [79]. As such, calf circumference measures may be used as a diagnostic proxy for older adults in settings where no other muscle mass diagnostic methods are available”.
The authors should reconsider this statement which may lead to misinterpretation.
8. LINES 202-205: “There is solid evidence of the four variables as independent risk factors for developing sarcopenia, functional decline, and dependence.”
Commentary: The assessment of muscle mass remains a challenge for international sarcopenia research groups(2, 3). On which grounds do the authors say that the evidence is solid?
9. LINES 215: “value above 4.76o can”…
Commentary: Change to 4.7
10. LINES 222-223: “In women, depression and the hormonal profile are some risk factors for sarcopenia, functional decline, and dependence.”…
Commentary: Insert a reference for this statement.
11. LINES 248-249: “Our work is the first including neural network analysis as its primary technique.”…
Commentary: Other studies (4) effectively use neural networks analysis (I quote just one). The authors might rephrase this sentence.
REFERENCES
1. Cruz-Jentoft AJ, Bahat G, Bauer J, Boirie Y, Bruyère O, Cederholm T, et al. Sarcopenia: revised European consensus on definition and diagnosis. Age Ageing. 2019;48(1):16-31.
2. Bhasin S, Travison TG, Manini TM, Patel S, Pencina KM, Fielding RA, et al. Sarcopenia Definition: The Position Statements of the Sarcopenia Definition and Outcomes Consortium. J Am Geriatr Soc. 2020;68(7):1410-8.
3. Cawthon PM, Manini T, Patel SM, Newman A, Travison T, Kiel DP, et al. Putative Cut-Points in Sarcopenia Components and Incident Adverse Health Outcomes: An SDOC Analysis. J Am Geriatr Soc. 2020;68(7):1429-37.
4. Yi, J., Shin, Y., Hahn, S. et al. Deep learning based sarcopenia prediction from shear-wave ultrasonographic elastography and gray scale ultrasonography of rectus femoris muscle. Sci Rep 12, 3596 (2022). https://doi.org/10.1038/s41598-022-07683-6
Reviewer 2 Report
This study aims to identify those patterns of muscle-related parameters and their association with sarcopenia in a cohort of older Mexican women, with neural network analysis. Although, no new information was provided.
-Abstract: is not concise with aim of this study.
-Methods: No reference was provided for questionnaires used.
-Results: This is a small sample size. In addition, regression analysis were not possible to do adequately.
Reviewer 3 Report
The aim of the study is to evaluate surrogate parameters to assess the the prevalence of sarcopenia in a sample of older Mexican women.
Sarcopenia was diagnosed according to the EWGSOP2 criteria.
Muscle related factors were calf circumference, phase angle from BIA measurement, gait speed and grip strength.
From a total of 449 participants 250 women were included. A flow chart is presented.
Analysis plan and ethic approval are presented.
Results: line 180 the authors say: " for the logistic regression analysis clusters 1,3 and 6 were eliminated because of small sample size. However, The sample size of the clusters is 11, 27, and 3. But, the sample size of cluster 5 is 15 and this cluster was included. Please explain this decision in detail.
Round 2
Reviewer 2 Report
No questions
Reviewer 3 Report
the authors addressed all comments